# Three-Month Durability of Bilateral Two-Level Stellate Ganglion Blocks for Traumatic Brain Injury: A Retrospective Analysis

**DOI:** 10.3390/biomedicines13071526

**Published:** 2025-06-23

**Authors:** Sean W. Mulvaney, Sanjay Mahadevan, Roosevelt J. Desronvilles, Kyle J. Dineen, Kristine L. Rae Olmsted

**Affiliations:** 1Department of Military and Emergency Medicine, Uniformed Services University, 4301 Jones Bridge Road, Bethesda, MD 20814, USA; seanmulvaney@hotmail.com; 2Orthobiologics Research Initiative Inc., 11200 Rockville Pike #230, North Bethesda, MD 20852, USA; roosevelt@rosm.org (R.J.D.J.); kyle@rosm.org (K.J.D.); 3RTI International, 3040 E Cornwallis Rd., Research Triangle Park, NC 27709, USA; krolmsted@rti.org

**Keywords:** stellate ganglion block, Traumatic Brain Injury, two-level cervical sympathetic chain block, 2LSCB, neurobehavioral symptoms, concussion

## Abstract

**Background/Objectives**: The primary aim of the study was to determine if ultrasound-guided, bilateral, two-level stellate ganglion blocks (SGBs), also known as two-level cervical sympathetic chain blocks (2LCSBs), performed on subsequent days, improve symptoms of chronic mild Traumatic Brain Injury (TBI) over a three-month period, as assessed by the Neurobehavioral Symptom Inventory (NSI). A secondary objective was to evaluate sex-based differences in outcomes during the same time period. **Methods**: A retrospective chart review was conducted between January 2024 and February 2025. We identified 41 patients who received bilateral 2LCSB for chronic (at least 3 months) TBI-related neurobehavioral symptoms as determined by NSI scores. NSI scores were collected at baseline, one week, one month, two months, and three months post treatment in 28 males and 13 females. An analysis of NSI scores and NSI-composite sub-scores was conducted to determine sex-based differences and 3-month differences in outcomes for patients receiving bilateral 2LCSB. **Results**: Of the 41 patients that underwent the bilateral 2LCSB procedure, 35 showed improvement in their NSI scores (85.36%) and 36 reported improvements in NSI sub-scores (87.8%). Across the entire dataset, patients experienced a 48.44% average decrease in total NSI scores from baseline and an average decrease of 43.11% in NSI sub-scores from baseline, indicative of improvements in TBI-specific symptoms. No statistical difference in outcomes was observed between males and females. **Conclusions**: Bilateral 2LCSB may provide rapid and durable TBI symptom improvement for 3 months, based on NSI scoring. However, additional research is necessary to establish causality.

## 1. Introduction

Traumatic Brain Injuries (TBIs) are a common medical injury and the leading cause of death and disability in children and adults, aged 1 to 44, with an average of over fifty thousand deaths and eighty thousand permanent disabilities each year [1]. The CDC reports the leading causes of TBI to be falls, motor vehicle accidents, assaults, and colliding with an object [2]. Moreover, recent studies suggest that up to 18.2% of the United States population has suffered a TBI, with evidence indicating that males are twice as likely to suffer a TBI than females [3] and that TBI is the second leading cause of disability in the country [4]. Additionally, brain trauma is a costly medical condition, both in care and lost productivity to society and is estimated at over USD 70 billion annually [5]. Military and athletic populations may be particularly vulnerable to TBIs. Approximately 28% of military members are known to have had one or more TBIs [6,7] and the reported prevalence of TBIs in college and high school athletes is increasing [8]. Of note, female athletes are 1.4 times as likely as males to suffer a TBI [9]. Furthermore, there is evidence to suggest that military-related TBI is associated with other neuropsychiatric conditions [10], as some research suggests that soldiers returning from combat deployment with a previous TBI had over a 40% incidence of comorbid PTSD [11].

TBIs are usually classified based on mechanism (penetrating or closed) and are further classified into three different categories based on the severity of injury: mild, moderate, or severe TBI [12]. Of note, these classifications are usually for acute injuries and are usually assessed based on the Glasgow Coma Scale (GCS) or Mayo classification system for Traumatic Brain Injury Severity. Both scales are used in clinical settings to characterize the severity of a TBI, enabling physicians to determine treatment options [13,14,15]. However, while effective in determining degree of pathology, the GCS lacks metrics to determine common persistent TBI-related symptoms in patients [16].

Closed-head TBIs are the most common category of TBI, and the term is sometimes used synonymously with the term “concussion” [17]. Blunt force to the head can cause damage to the axon cords of neurons and may impact cerebral blood flow [18]. Studies suggest that microtubules are the weak link of the axonal cytoskeleton and are most likely to be damaged due to the rapid stretching of these structures that occurs during a concussion [19,20]. Furthermore, microtubule damage as sequelae of TBI or concussions is suspected to cause tau protein accumulation through prion-like spread and is associated with other neurodegenerative conditions like chronic traumatic encephalopathy and Alzheimer’s disease [21,22,23,24].

Over 90% of TBIs are considered mild and patients typically experience neurobehavioral symptom changes associated with post-concussion syndrome [25]. Additionally, for patients with possible TBI or post-concussive syndrome, neurobehavioral symptoms may persist for an extended period and are considered persistent if they remain present beyond three months [26].

There appears to be a gap in the treatment of chronic sequelae of mild to moderate TBIs. Current treatment options to address mild to moderate TBI include medication [27,28], physical and manual therapies [29], hyperbaric oxygen therapy (HBOT) [30,31,32], cognitive behavioral therapy [33], and transcranial stimulation [34]. While some of these interventions have shown promise, most clinicians agree that treating patients with TBIs requires individualization as opposed to homogeneity [35].

Recently, the stellate ganglion block (SGB) has emerged as a potential therapeutic intervention that can aid in resolving neurobehavioral symptoms after a TBI. The SGB procedure has most commonly been used to treat posttraumatic stress disorder (PTSD) and was first used in 1925 to treat posttraumatic stress injury [36]. The procedure was later described in a 1990 case report by Lebovits et al. in a patient with symptoms of PTSD who was being treated for a chronic pain syndrome; the patient’s pain and PTSD symptoms improved [37]. Since then, a number of publications evaluating SGB for the treatment of PTSD have appeared in the literature [38,39,40,41]. In addition, there is research supporting the efficacy of SGB for the treatment of PTSD, as reported in a multi-center randomized controlled trial (RCT) [42]. Finally, the procedure has been shown to be well accepted by patients [43].

Anatomically, the stellate ganglion is a fusion of the inferior cervical and first thoracic sympathetic ganglia that run along the anterior cervical trunk at C7 cervical vertebrae. These ganglia are hypothesized to modulate signaling of the sympathetic nervous system as a part of the cervical sympathetic trunk [44]. Of note, approximately 20% of patients are found to not have a stellate ganglion [45,46]. There is significant anatomic variation in the course of the cervical sympathetic chain [45]. Although we endorse the term 2LCSB, this term is not currently widely acknowledged. The use of the term 2LCSB may limit the ability of clinicians and researchers to locate these studies in the body of medical literature; therefore, we acknowledge the term SGB but continue to refer to 2LCSB throughout the rest of this study. A series of publications have since established that bilateral blocks provided superior outcomes to single-sided blocks [47,48,49,50].

A prior study showed that in addition to improvements in PTSD and anxiety symptoms, certain populations (such as military veterans with a history of TBI) reported improvements in neurobehavioral symptoms one month after 2LCSB [51]. To more accurately assess these symptoms, a subsequent study included the Neurobehavioral Symptom Inventory (NSI) [52] as an outcome measure as part of standard initial screening and follow-up and found that neurobehavioral symptoms not associated with PTSD improved in patients with a history of TBI one month after 2LCSB [53]. While the NSI does not diagnose TBI, it is recognized to have value as a self-reporting tool for patients and clinicians to assess neurobehavioral symptom changes [54] and has been adopted by the U.S. Department of Defense and the Department of Veterans Affairs for both research and clinical evaluation of TBI [55]. Moreover, as a self-report tool, the NSI tracks symptoms not captured by the GCS or Mayo classification system.

Despite emerging evidence for the use of 2LCSB for neuropsychiatric disorders, including TBI, no studies have been published to date that evaluate the durability of 2LCSB for improving TBI symptoms beyond one month. The primary purpose of this study is to evaluate the durability of 2LCSB for TBI over a period of three months, with a secondary aim of examining differences in outcomes based on sex.

## 2. Materials and Methods

This retrospective case series was approved by the institutional review board of the Institute of Regenerative and Cellular Medicine (ICRM-2024-413) and included 41 patients (28 male and 13 female) that received bilateral 2LCSB between January 2024 and January 2025. Patients had either self-identified with having previously suffered a TBI or were previously diagnosed with TBI by a clinician.

Patients were included based on the following criteria: a health provider’s formal assessment of a history of TBI-related neurobehavioral symptoms; TBI-related symptom duration of at least 3 months; physician evaluation; and NSI score. Because the data reported here are from a clinical practice, there were no exclusion criteria. The items on the NSI are scored on a scale of 0 to 4, with 22 items and a range of 0 to 88. The instrument was provided to patients through a secure email system, via paper copy, or via a cloud-based, 21 CFR Part 11 compliant, secure data collection system. Patients who completed the NSI at all time points (baseline, 1 week, 1 month, 2 months, and 3 months relative to procedure time point) were included in the analytic dataset.

The clinical and procedural details of the 2LCSB technique used in this study have been previously published [48]. Briefly, ultrasound guidance was used to introduce 6–8 milliliters of 0.5% ropivacaine at the 6th cervical vertebra and 1.5–2 milliliters of 0.5% ropivacaine at the 4th cervical vertebra on the right side. The procedure was repeated on the left side on the following day to eliminate the risk of inadvertent bilateral blockade of the recurrent laryngeal nerve and subsequent potential airway compromise [49]. All procedures were performed by a single fellowship-trained physician who has performed over 4000 2LCSBs.

For this study, we identified symptoms assessed by the NSI that were not assessed by the PCL-5. A sub-score for these non-overlapping questions was generated, with a possible range of 0 to 40 points. These questions covered the following: symptomatic history of dizziness, loss of balance and coordination, headaches, nausea, vision problems, hearing difficulty, sensitivity to light and noise, loss of smell and taste, and numbness or tingling in areas of the body. This sub-score was used to determine if any improvements associated with 2LCSB seen may have been due to improvements in TBI-related neurobehavioral symptoms versus improvements in other comorbid conditions such as anxiety or PTSD.

We conducted a retrospective analysis of NSI scores and the NSI-unique sub-scores from baseline to three months to determine changes in patient symptoms. The outcomes were separated and analyzed to determine sex-based differences and changes in scores by time point. For our primary objective, to test if there was a difference in change in NSI scores and NSI sub-scores from baseline to three months, a paired *t*-test was used to examine statistical significance. For our secondary objective, to test if there was a statistically significant difference in outcomes for males and females at three months, an independent samples *t*-test was used.

## 3. Results

In total, 35 of the 41 patients (85.36%) treated with bilateral 2LCSB in our retrospective analysis saw a decrease in NSI scores, and 36 out of 41 patients reported a decrease in NSI sub-scores (87.8%) between baseline and the three-month follow-up. The average age for included patients was 43.6 years (46.4 for males and 38.2 for females). More than two-thirds of participants were male (n = 28). There was no difference between male and female NSI and NSI sub-scores at baseline (pre-treatment). To test our primary hypothesis, we conducted a paired *t*-test for both change in NSI scores and change in NSI sub-scores over three months. Regarding our first hypothesis, patients reported significant decreases in NSI score after three months (mean change {SD, [95% CI]}, 19.37 {17.18, [13.76, 24.97]}; t (40) = 6.98, *p* =< 0.001). Similarly, patients reported significant decreases in NSI sub-score after three months (mean change {SD, [95% CI]}, 5.92 {7.7, [3.58, 8.26]}; t (40) = 5.09, *p* =< 0.001).

To test our secondary hypothesis, we conducted two independent samples *t*-tests to examine if there was a difference in outcomes between male and female patients at 3 months post 2LCSB. The mean (SD) NSI score at 3 months was 20.65 (16.3): 19.21 (15.88) for males and 23.62 (19.21) for females. This difference was not statistically significant, and the effect size was small (t(unequal variances) = −0.72, *p* = 0.48, Cohen’s *d* = −0.26). The mean (SD) NSI sub-score at 3 months was 7.86 (6.73): 7.58 (6.48) for males and 8.4 (7.4) for females. Again, there was no significant difference between males and female on NSI sub-scores at 3 months, and the effect size was small (t(unequal variances) = −0.36, *p* = 0.72, Cohen’s *d* = −0.12).

Finally, regarding our second hypothesis, we conducted two additional independent sample *t*-tests to examine if there was a difference between male and female patients in improvements in NSI score and NSI sub-score. The mean (SD) change in NSI score at 3 months was 19.37 (17.18): 22.93 (18.89) for males and 11.54 (12.31) for females. Based on this, there was a significant difference in the change in NSI score with a medium-to-large effect size (t = 2.32, *p* = 0.026, Cohen’s *d* = 0.67). The mean (SD) change in NSI sub-score at 3 months was 5.92 (7.7): 7.07 (8.47) for males and 3.7 (5.55) for females. Although, males showed a slightly higher average improvement, there is no significant difference between males and females in change in NSI sub-score at 3 months. Results and descriptive statistics are presented in Figure 1, Figure 2, Figure 3 and Figure 4 and Table 1. 

## 4. Discussion

The primary aim of this study is to assess the durability of 2LCSB over three months in patients with a history of TBI. While there is support for 2LCSB for TBI symptom improvement at one month [51,53], there are no studies which have evaluated longer-term changes. The results of this study support the original hypothesis that bilateral 2LCSB conveys durable improvements in chronic symptom sequelae of TBI up to three months. As previously mentioned, HBOT has emerged as a potential therapy to treat patients suffering from chronic neurobehavioral symptoms after a TBI [56], with an RCT demonstrating an average decrease in NSI score of 10.6 points after three months. The results of this study demonstrate an average decrease in NSI score of nearly double that in the HBOT RCT (19.37 points). However, these results must be interpreted with caution due to differences in design between the two studies. Further research should be conducted to examine head-to-head differences between the two interventions for the treatment of TBI.

The mechanism by which 2LCSB affects TBI-related neurobehavioral symptoms is not currently well understood. Some evidence points to 2LCSB improving neural perfusion patterns in the cerebral hemisphere, modulation of neuroinflammatory factors (such as NF-kB), and regulation of nerve–endocrine–immune system signaling to contribute to improved symptoms [57,58,59]. Some evidence also suggests that TBI is a chronic disease marked by persistent neural inflammation in part due to microglial cell dysfunction [60], pro-inflammatory cytokines [61], and plasma protein dysfunction [62], among many other influencing factors.

Extant literature suggests that the mechanical impact of TBI and initial swelling can cause immediate hypoxia and disrupted cerebral blood flow. These, in turn, can lead to the development of chronic pathological cellular processes such as persistent neuroinflammation [63]. In the subacute to chronic phase of the injury, translational research suggests that continued blood–brain barrier (BBB) disruption, mitochondrial dysfunction, and chronic activation of microglia and astrocytes can contribute to persistent neuroinflammation [64,65]. In response to hypoxic conditions, microglia are signaled to alter morphology, increasing motility towards injury sites, and enhancing phagocytosis to clear damaged cells and debris [66]. Microglia can alter their phenotype depending on the severity and duration of brain injury. Acute hypoxia can induce an M2 anti-inflammatory, neuroprotective state, while chronic hypoxia often drives an M1 neurotoxic state, releasing pro-inflammatory cytokines, such as TNF-α, IL-1β, and IL-6, and reactive oxygen species [64,67]. These pro-inflammatory cytokines released by activated microglia stimulate astrocytes to proliferate and produce additional inflammatory mediators, thereby amplifying neuroinflammation. While astrocytosis initially occurs to protect the central nervous system, excessive or chronic astrocytosis can impair normal astrocyte functions, disrupt BBB integrity, alter neurotransmitter regulation, and contribute to synaptic dysfunction and cognitive impairment [68]. Other translational research has demonstrated that progressive gray and white matter atrophy is observed after a TBI [69]. Some studies have also linked the presence of pro-inflammatory cytokines in biofluid (blood or CSF) following a TBI to persistent neurobehavioral symptoms [70,71]. Finally, research has documented the presence of neurobehavioral symptoms after a TBI up to or greater than three months after injury [72,73]. While the findings of this study do not directly address the presence of biofluid biomarkers or cerebral perfusion patterns, they do support an improvement in neurobehavioral symptoms up to 3 months after a TBI following 2LCSB. Although there is not currently a consensus on how bilateral 2LCSB might improve symptoms, either for TBI or other neuropsychiatric disorders, there is evidence that shows that 2LCSB enhances cerebral blood flow using transcranial Doppler as well as cerebral perfusion patterns on neuroimaging [74,75,76,77]. In chronic TBI, there are persistent areas of decreasing cerebral perfusion [78,79]. We hypothesize that the improvements in cerebral perfusion after bilateral 2LCSB could be a mechanism for the symptom improvements observed, in part by reversing the hypoxia-driven microglial cell activation, which in turn drives the chronic astrocytosis [68]. Some human studies suggest that the chronic increase in sympathetic tone in TBI may be detrimental to the function of the BBB [80,81]. By decreasing the chronic sympathetic tone with 2LCSB, we may be improving the integrity of the BBB, resulting in decreased inflammatory peripheral immune cells entering the brain. This may, in turn, contribute to some of the durable symptom improvements observed here. Further studies are needed to fully evaluate and test this hypothesis, especially in the indication of TBI and other neuropsychiatric disorders.

The secondary aim of this study is to assess potential sex-based differences in outcomes of 2LCSB in patients who had a history of TBI. Based on our data, we do not have a hypothesis for the discrepancy in change in NSI score between male and female patients. As previously mentioned, patients who suffer a TBI might also be at risk for developing comorbid neuropsychiatric disorders [11]. Despite the difference between male and female change in NSI score and NSI sub-score, we find the improvement in NSI sub-score to be clinically valuable since this composite score intends to more specifically characterize neurobehavioral symptoms as opposed to the overall NSI score. This is validated by similar NSI scores and NSI sub-scores in male and female patients at three months. These findings are in accordance with those from previous studies [51,53]. As such, 2LCSB may be considered regardless of sex when persistent TBI pathology is present.

There are significant limitations to this study. As it is a retrospective analysis of clinical data, the lack of a control arm limits the generalizability of the findings as well as the strength of the conclusions that can be drawn. Also, the symptom improvements reported in this study may be due to placebo, which has been reported to be as high as 50% with injection procedures [82]. Other minimally invasive musculoskeletal procedures can produce significant placebo effects [83]. Future RCTs that implement an appropriate randomization style and a sham injection control arm can, at least in part, address these concerns. Another limitation is the reliance of our findings on self-reported TBI symptoms. Self-reporting in TBI is limited by cognitive deficits associated with the injury, the stigma associated with both cognitive deficits and behavioral issues, and unclear terminology related to TBI (concussion, knocked out, etc.) [84,85,86]. Finally, a more comprehensive assessment of patient history regarding symptoms, including prior TBI diagnosis, more accurate duration of time since injury, employment status, and related pathologies (i.e., dysautonomia, cervical injury, comorbid neuropsychiatric disorders), may improve both the accuracy of study results and clinical utility.

## 5. Conclusions

Chronic and acute mild to moderate TBI is a condition that currently has limited effective treatment options. The findings of this retrospective study suggest that bilateral 2LCSB may result in rapid and durable symptom relief for at least 3 months and may be an effective treatment. Randomized controlled trials are needed to better establish the effectiveness of the procedure. Such studies may also suggest a mechanism of action, or validate the mechanism proposed here, thus enabling more targeted interventions for mild to moderate TBI.

## Figures and Tables

**Figure 1 biomedicines-13-01526-f001:**
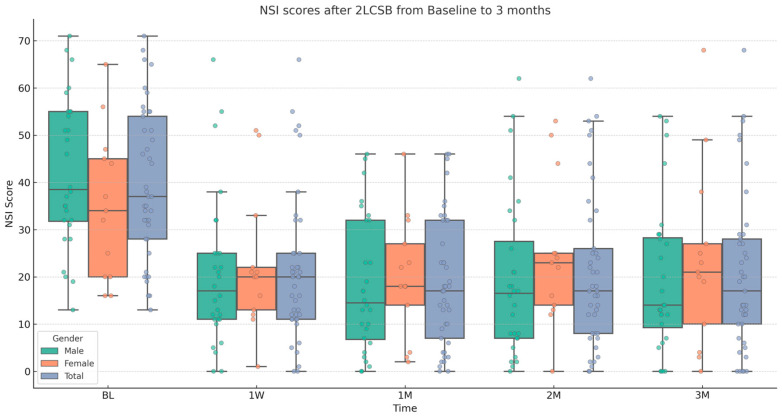
Box and whisker plot, depicting NSI scores for male, female, and all patients at baseline (BL), 1 week (1W), 1 month (1M), 2 months (2M), and 3 months (3M) after bilateral, two-level cervical sympathetic blockade (2LCSB). The middle line of each box at each time point represents the median NSI score. The top of each box represents the 75th percentile (Q3) NSI score. The bottom of each box represents the 25th percentile (Q1) NSI score. The height of each box represents the Interquartile Range (IQR), or Q3–Q1. The top whisker is the highest value NSI score within 1.5 times the IQR. The bottom whisker is the lowest value NSI score within 1.5 times the IQR. This box plot is overlayed on a dot plot of the raw individual data points to visualize the density of the NSI scores.

**Figure 2 biomedicines-13-01526-f002:**
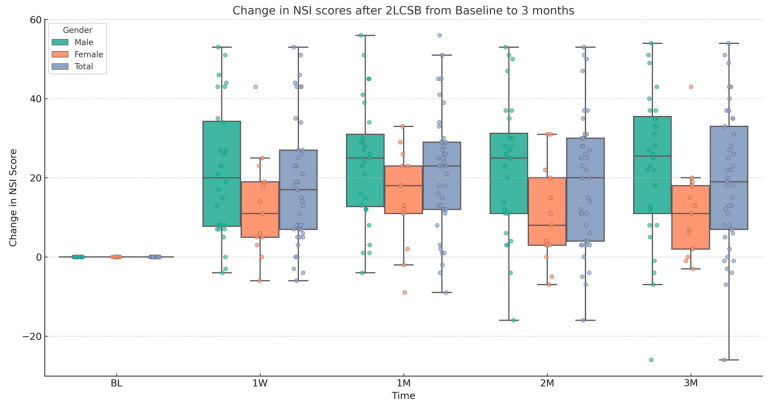
Box and whisker plot, depicting the change in NSI scores for male, female, and all patients at baseline (BL), 1 week (1W), 1 month (1M), 2 months (2M), and 3 months (3M) after bilateral, two-level cervical sympathetic blockade (2LCSB). The middle line of each box at each time point represents the median change in NSI score. The top of each box represents the 75th percentile (Q3) change in NSI score. The bottom of each box represents the 25th percentile (Q1) change in NSI score. The height of each box represents the Interquartile Range (IQR), or Q3–Q1. The top whisker is the highest value change in NSI score within 1.5 times the IQR. The bottom whisker is the lowest value change in NSI score within 1.5 times the IQR. This box plot is overlayed on a dot plot of the raw individual data points to visualize the density of the change in NSI scores.

**Figure 3 biomedicines-13-01526-f003:**
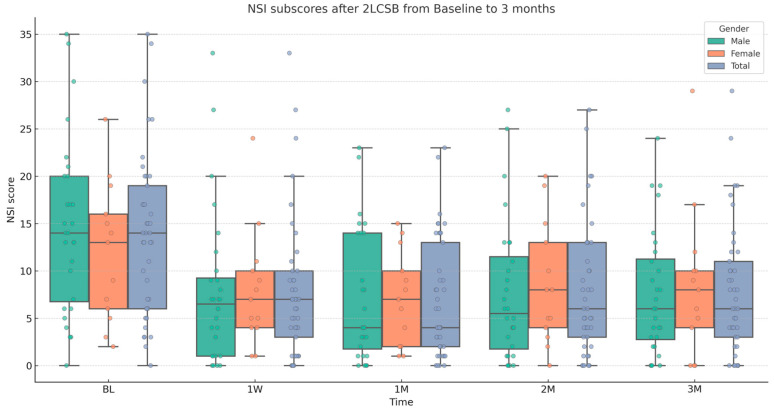
Box and whisker plot, depicting NSI sub-scores for male, female, and all patients at baseline (BL), 1 week (1W), 1 month (1M), 2 months (2M), and 3 months (3M) after bilateral, two-level cervical sympathetic blockade (2LCSB). The middle line of each box at each time point represents the median NSI sub-score. The top of each box represents the 75th percentile (Q3) NSI sub-score. The bottom of each box represents the 25th percentile (Q1) NSI sub-score. The height of each box represents the Interquartile Range (IQR), or Q3–Q1. The top whisker is the highest value NSI sub-score within 1.5 times the IQR. The bottom whisker is the lowest value NSI sub-score within 1.5 times the IQR. This box plot is overlayed on a dot plot of the raw individual data points to visualize the density of the NSI sub-scores.

**Figure 4 biomedicines-13-01526-f004:**
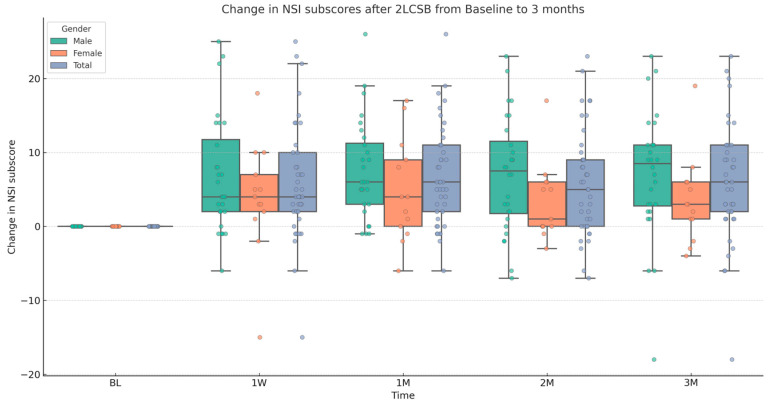
Box and whisker plot, depicting the change in NSI sub-scores for male, female, and all patients at baseline (BL), 1 week (1W), 1 month (1M), 2 months (2M), and 3 months (3M) after bilateral, two-level cervical sympathetic blockade (2LCSB). The middle line of each box at each time point represents the median change in NSI sub-score. The top of each box represents the 75th percentile (Q3) change in NSI sub-score. The bottom of each box represents the 25th percentile (Q1) change in NSI sub-score. The height of each box represents the Interquartile Range (IQR), or Q3–Q1. The top whisker is the highest value change in NSI sub-score within 1.5 times the IQR. The bottom whisker is the lowest value change in NSI sub-score within 1.5 times the IQR. This box plot is overlayed on a dot plot of the raw individual data points to visualize the density of the change in NSI sub-scores.

**Table 1 biomedicines-13-01526-t001:** Average NSI scores, NSI sub-scores, change in NSI scores, and change in NSI sub-scores for male, female, and total patients, at baseline and three months.

Average NSI and NSI Sub-Scores	Male (SD)	Female (SD)	Total (SD)
Count	28	13	41
Baseline NSI scores	42.21 (15.87)	35.15 (15.68)	39.92 (15.42)
Three-month NSI scores	19.21 (15.88)	23.62 (19.21)	20.65 (16.3)
Avg. Change in NSI Score (BL to 3M)	22.93 (18.89)	11.54 (12.31)	19.37 (17.18)
Percent Decrease from Baseline	54.16%	32.82%	48.44%
Baseline NSI sub-scores	14.65 (9.0)	12.1 (6.79)	13.78 (8.32)
Three-month NSI sub-scores	7.58 (6.48)	8.4 (7.4)	7.86 (6.73)
Avg. Change in NSI sub-score (BL to 3M)	7.07 (8.47)	3.7 (5.55)	5.92 (7.7)
Percent Decrease from Baseline	48.22%	29.03%	43.11%

## Data Availability

The data that support the findings of this study are available from the corresponding author on request due to privacy issues in accordance with the consent provided by the participants. All data are freely accessible.

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
