# Peer review of "Three-Month Durability of Bilateral Two-Level Stellate Ganglion Blocks for Traumatic Brain Injury: A Retrospective Analysis"

_biomedicines, 2025, doi:10.3390/biomedicines13071526_

Round 1

Reviewer 1 Report

Comments and Suggestions for Authors

In this manuscript, Sean W et al. introduce a two-level stellate ganglion blocks (SGB), which can improve symptoms of chronic mild Traumatic Brain Injury (TBI) through three months, as assessed by the Neurobehavioral Symptom Inventory (NSI). The method of SGB in the treatment of TBI is interesting. However, there are significant problems in the manuscript that need to be addressed.

1) The author has repeatedly emphasized in the article that the NSI score is an assessment indicator for evaluating the improvement of patients after TBI, but has not specified the evaluation method of the NSI score in the article. Please specify the evaluation method of the NSI score in this article.

2) The author adopts NSI score and NSI sub-score respectively in Figures 1-4 in the text. Why these two quantitative indicators are adopted simultaneously? Please elaborate in detail in the text by the author.

3) In the article, the author presented the NSI and NSI sub-scores of BL, 1 week, 1month, 2 months, and 3 months in TBI patients who received LCSB. Does the author have the patient data after 3 months? If so, please display it in the text. If not, please explain why it is only up to 3 months.

4) Although the author put forward the viewpoint on the gender-based outcome differences of TBI patients who received LCSB during the same period, the current discussion on different factors of patients is not rich enough. It is suggested that the author increase the research on the differences of different factors such as age in future studies.

5) Since an important part of the manuscript is about the comparison of LCSB between different genders at the same time, the comparison factor of "gender" needs to be added in the title of this article.

6) We suggested that author conduct animal experiments to explore the related mechanism research of the improvement effect of LCSB on patients with TBI in future research.

7) Some abbreviations of professional terms in the text do not appear in the abbreviations on page 9, such as Interquartile Range (IQR) on page 6. Author should complete the abbreviations.

8) The evidence proposed by the author on page 8 of the article that LCSB improves the integrity of BBB by reducing the chronic sympathetic nerve of 2LCSB, thereby reducing the entry of inflammatory peripheral immune cells into the brain, is not  sufficient and requires further verification and confirmation through subsequent experiments.

Author Response

  1. The author has repeatedly emphasized in the article that the NSI score is an assessment indicator for evaluating the improvement of patients after TBI, but has not specified the evaluation method of the NSI score in the article. Please specify the evaluation method of the NSI score in this article.
    1. We thank the reviewer for this comment. On lines 153-157, we detail that a paired t-test was used to evaluate statistical differences in outcomes from BL to 3M for all patients and an independent samples t-test was used to examine differences in sex-based outcomes. As such, no revisions have been made.
  2. The author adopts NSI score and NSI sub-score respectively in Figures 1-4 in the text. Why these two quantitative indicators are adopted simultaneously? Please elaborate in detail in the text by the author.
    1. Per lines 142-149, we chose to include this NSI sub-score to assess TBI-related neurobehavioral symptoms that were not assessed by the PCL-5. Moreover, this sub-score was used to determine if any improvements associated with 2LCSB seen may have been due to improvements in TBI-related neurobehavioral symptoms versus improvements in other comorbid conditions such as anxiety or PTSD. We believe presentation of both is important for readers to understand the findings relative to one another. No revisions have been made.
  3. In the article, the author presented the NSI and NSI sub-scores of BL, 1 week, 1month, 2 months, and 3 months in TBI patients who received LCSB. Does the author have the patient data after 3 months? If so, please display it in the text. If not, please explain why it is only up to 3 months.
    1. At the time of submission, only the data out to 3 months were available. Per lines 306-308 and lines 325-328, this is a preliminary retrospective study of clinical outcomes upon receiving the 2LCSB treatment, so we believe a relatively shorter follow-up period is appropriate. In addition, per reference 58, the HBOT RCT that we reference in the manuscript used 13 weeks as a preliminary timepoint to assess effectiveness (line 241), and we wished to mirror that time frame for comparison purposes. No revisions have been made to the manuscript.
  4. Although the author put forward the viewpoint on the gender-based outcome differences of TBI patients who received LCSB during the same period, the current discussion on different factors of patients is not rich enough. It is suggested that the author increase the research on the differences of different factors such as age in future studies.
    1. We appreciate this comment from the reviewer and agree that future research should analyze outcomes of 2LCSB for TBI by age and other factors that might contribute to the symptomatic response from the intervention.
  5. Since an important part of the manuscript is about the comparison of LCSB between different genders at the same time, the comparison factor of "gender" needs to be added in the title of this article.
    1. As noted in lines 16-17, 117-118, and elsewhere, our analysis of outcomes based on sex was secondary. As such, we have chosen not to include sex in the title.
  6. We suggested that author conduct animal experiments to explore the related mechanism research of the improvement effect of LCSB on patients with TBI in future research.
    1. We appreciate this comment from the reviewer and agree that future animal studies regarding mechanism may be warranted.
  7. Some abbreviations of professional terms in the text do not appear in the abbreviations on page 9, such as Interquartile Range (IQR) on page 6. Author should complete the abbreviations.
    1. We thank the reviewer for this comment and have added appropriate abbreviations in the table at the bottom of the manuscript.
  8. The evidence proposed by the author on page 8 of the article that LCSB improves the integrity of BBB by reducing the chronic sympathetic nerve of 2LCSB, thereby reducing the entry of inflammatory peripheral immune cells into the brain, is not sufficient and requires further verification and confirmation through subsequent experiments.
    1. Per line 285, we propose this as a potential explanatory mechanism but acknowledge the need for future research to confirm this hypothesis. No revisions to the manuscript have been made.

Reviewer 2 Report

Comments and Suggestions for Authors

Mulvaney et al. investigated the durability of the stellate ganglion block for TBI.

Can the authors elaborate why there are no exclusion criteria?

Why was NSI chosen to be evaluated?

How can this study be a retrospective study? The reviewer is assuming that the authors were already planning in to do the manuscript, and this is the reason why the patient had prior scores recorded.

Did all the patients had the same characteristics only TBI without any other medical problem and not taking any other medication? Please address this in the limitations.

How can this study not be considered a clinical trial?

How did the authors explain to the patients what was being performed?

How was calculated the power of the study?

What was the statistical software used in the study?

How did the authors evaluate the effect of confounding variables?

Author Response

  1. Can the authors elaborate why there are no exclusion criteria?
    1. We thank the reviewer for this comment. The retrospective data reported here were collected in the course of clinical practice from patients sought treatment for their condition. Only data from patients deemed by the provider to be appropriate candidates for treatment were included. Thus, there were no distinct exclusion criteria for this study.
  2. Why was NSI chosen to be evaluated?
    1. The NSI is frequently used to characterize change in symptoms over time. Per Line 110 and reference 54, we chose the NSI because it is recognized to have value as a self-reporting tool for patients and clinicians to assess neurobehavioral symptom changes. Additionally, the HBOT study referenced on lines 240-244 (reference 58) also used the NSI as an endpoint to assess symptom change over time.
  3. How can this study be a retrospective study? The reviewer is assuming that the authors were already planning in to do the manuscript, and this is the reason why the patient had prior scores recorded.
    1. The current IRB approved study is retrospective in nature because it represents observational review of historical clinical patient records. This definition can be seen in de Sanctis et al. 2022 as well as other sources (https://pmc.ncbi.nlm.nih.gov/articles/PMC9686178/).
  4. Did all the patients had the same characteristics only TBI without any other medical problem and not taking any other medication? Please address this in the limitations.
    1. As these retrospective data were collected in the usual course of clinical care based on patients who sought treatment for their condition, the authors did not have full access to patient records. As such we did not have access to information regarding other potential comorbidities and medications. No revisions to the manuscript have been made.
  5. How can this study not be considered a clinical trial?
    1. Clinical trials are prospective in nature and involve assignment to one or more interventions, as can be seen at (https://grants.nih.gov/policy-and-compliance/policy-topics/clinical-trials/definition) as well as in other references. Our IRB approved study included data collected as part of standard clinical care. We have presented a retrospective case series in which no assignment to interventions took place.
  6. How did the authors explain to the patients what was being performed?
    1. Per the informed consent provided by the authors at the time of submission, the treating physician discussed the intervention in person with patients/participants, as well as potential risks and benefits of receiving the treatment.
  7. How was calculated the power of the study?
    1. Since the results of this study were based on novel clinical retrospective data via chart review, power analysis is not appropriate.
  8. What was the statistical software used in the study?
    1. We used JASP to generate the results of the statistical tests. JASP Team (2024). JASP (Version 0.19.3)[Computer software].
  9. How did the authors evaluate the effect of confounding variables?
    1. As this is a retrospective case series, no controlling for confounders is possible. Had this been an experimental design study with selection criteria, we agree evaluation of confounding variables would be important.

Round 2

Reviewer 2 Report

Comments and Suggestions for Authors

Satisfactory